# Growth Regulators Improve Outcrossing Rate of Diverse Rice Cytoplasmic Male Sterile Lines through Affecting Floral Traits

**DOI:** 10.3390/plants11101291

**Published:** 2022-05-12

**Authors:** Essam A. Z. ElShamey, Hassan Sh. Hamad, Khalid S. Alshallash, Mousa A. Alghuthaymi, Mohamed I. Ghazy, Raghda M. Sakran, Mahmoud E. Selim, Mahmoud A. A. ElSayed, Taher M. Abdelmegeed, Salah A. Okasha, Said I. Behiry, Ridha Boudiar, Elsayed Mansour

**Affiliations:** 1Rice Research and Training Center, Field Crops Research Institute, Agricultural Research Center, Kafrelsheikh 33717, Egypt; essamelshamey@gmail.com (E.A.Z.E.); hassanshehata28@yahoo.com (H.S.H.); m_ghazy2050@yahoo.com (M.I.G.); raghdasakran@yahoo.co.uk (R.M.S.); m.selimrrtc@gmail.com (M.E.S.); mahmoud.rrtc@gmail.com (M.A.A.E.); taher.mohamed2018@gmail.com (T.M.A.); 2College of Science and Humanities-Huraymila, Imam Mohammed Bin Saud Islamic University (IMSIU), Riyadh 11432, Saudi Arabia; 3Biology Department, Science and Humanities College, Shaqra University, Alquwayiyah 11726, Saudi Arabia; 4Department of Agronomy, Faculty of Agriculture, Suez Canal University, Ismailia 41522, Egypt; salah.okasha@agr.suez.edu.eg; 5Agricultural Botany Department, Faculty of Agriculture (Saba Basha), Alexandria University, Alexandria 21531, Egypt; said.behiry@alexu.edu.eg; 6Division of Biotechnology & Agriculture, Biotechnology Research Center (CRBt), UV 03, P.O. Box E73, Nouvelle Ville Ali Mendjli, Constantine 25016, Algeria; boudiarreda@yahoo.fr; 7Department of Crop Science, Faculty of Agriculture, Zagazig University, Zagazig 44519, Egypt

**Keywords:** foliar application, growth regulator, CMS line, hybrid rice, floral traits, grain yield, principal component analysis

## Abstract

Cytoplasmic male sterility (CMS) provides an irreplaceable strategy for commercial exploitation of heterosis and producing high-yielding hybrid rice. The exogenous application of plant growth regulators could improve outcrossing rates of the CMS lines by affecting floral traits and accordingly increase hybrid rice seed production. The present study aimed at exploring the impact of growth regulators such as gibberellic acid (GA_3_), indole-3-acetic acid (IAA), and naphthalene acetic acid (NAA) on promoting floral traits and outcrossing rates in diverse rice CMS lines and improving hybrid rice seed production. The impact of foliar applications of growth regulators comprising GA_3_ at 300 g/ha or GA_3_ at 150 g/ha + IAA at 50 g/ha + NAA at 200 g/ha versus untreated control was investigated on floral, growth, and yield traits of five diverse CMS lines. The exogenously sprayed growth regulators, in particular, the combination of GA_3_, IAA, and NAA (T3) boosted all studied floral, growth, and yield traits in all tested CMS lines. Moreover, the evaluated CMS lines exhibited significant differences in all measured floral traits. L2, L3, and L1 displayed the uppermost spikelet opening angle, duration of spikelet opening, total stigma length, style length, stigma brush, and stigma width. In addition, these CMS lines exhibited the highest plant growth and yield traits, particularly under T3. Consequently, exogenous application of GA_3_, IAA, and NAA could be exploited to improve the floral, growth, and yield traits of promising CMS lines such as L2, L3, and L1, hence increasing outcrossing rates and hybrid rice seed production.

## 1. Introduction

Rice (*Oryza sativa*) is a significant food crop and the primary source of nutrition for more than half of the world’s population [1]. Its total cultivated area is around 165 million hectares producing about 757 million tons yearly [2]. Rice production demand keeps increasing due to the abrupt global climate change and fast-growing global population [3]. The exploitation of F1 hybrid vigor in rice is one of the important approaches to enhancing its production and food security worldwide [4]. Hybrid rice provides markedly greater production and net return compared to conventional varieties [5,6]. Notwithstanding, the seed production of rice hybrid is limited due to it being a self-pollinated crop and requires a cytoplasmic male sterile (CMS) line, a maintainer (B) line, and a restorer (R) line [7]. CMS is used as the female parent, which is propagated via cross-pollination by the corresponding B line, and the R line is used to pollinate the CMS line for producing male-recovered hybrid rice seeds with heterosis [8]. The drawback of the three-line method is the low seed production, mainly due to poor panicle exertion and the low outcrossing rate [9]. 

Enhancing outcrossing rates in the CMS lines considerably increases hybrid seed production [10]. Plant growth regulators could be used as exogenous applications to enhance hybrid rice seed production [11]. Several earlier reports elucidated that the exogenous application of plant growth regulators such as gibberellic acid (GA_3_), improved hybrid rice seed production. In this context, Tiwari et al. [12] demonstrated that GA_3_ followed by NAA, urea, and K_2_PO_4_ enhanced hybrid rice seed production. Similarly, Pan et al. [13] deduced that spraying 50 mg/L paclobutrazol followed by 30 mg/L 6-benzylaminopurine and gibberellic acid at the heading stage increased the number of spikelets/panicle, seed setting rate, and rice grain yield. Furthermore, Soomro et al. [14] depicted that foliar application of GA_3_ followed by NAA boosted plant height, number of tillers, grain filling percentage, and grain yield of rice. Thus, plant growth regulators demonstrate a considerable impact in enhancing plant development, plant growth, and productivity, but little is known about their impacts on floral traits. Accordingly, more knowledge is still needed about the impact of plant growth regulators on floral traits and its role in improving hybrid rice seed production.

The regulation of floral traits is vitally decisive in hybrid rice seed production. Plant growth regulators have positive impacts on panicle exertion, stigma exertion, duration of floret opening, panicle exertion from the flag leaf sheath, and seed setting rate of A-lines [15]. GA_3_ is a plant growth hormone that promotes cell elongation and has a substantial impact on plant development [16]. Its foliar application at the heading stage enhances floret opening duration, stigma properties, panicle exertion of male sterile lines, and rate of stigma exertion and improves out-crossing rate and seed yield potential during hybrid rice seed production [11]. Likewise, indole-3-acetic acid (IAA) is a vital hormone for several aspects of plant physiology such as cell division, cell differentiation, cell elongation, protein synthesis, and chlorophyll content [17,18]. IAA stimulates root formation, protects photosynthetic machinery, delays senescence, elevates fruit ripening, and enhances pathogen–plant interaction [19]. Moreover, naphthalene acetic (NAA) acid promotes root formation, cell division, tissue swelling, callus initiation, embryogenesis induction, photosynthetic efficiency, and floral characteristics [20,21,22]. Accordingly, the combination of these promising regulators could enhance floral traits and improve hybrid rice seed production. Thereupon, the present study aimed at exploring the influence of gibberellic acid, indole-3-acetic acid, and naphthalene acetic acid on floral traits and hybrid rice seed production. Moreover, to assess the performance of diverse rice cytoplasmic male sterile lines under the application of used growth regulators. 

## 2. Results

### 2.1. Floral Traits

The application of growth regulators, CMS lines, and their interaction displayed highly significant (<0.001) effects on all evaluated floral traits (Table 1). The application of T3 with combination of GA_3_, IAA, and NAA, reflected the greatest enhancement of all floral traits compared to untreated control T1 (Table 1 and Appendix A). The application T3 enhanced spikelet opening angle (SOA) by 27.7, duration of spikelet opening (DSO) by 16.9%, total stigma length (TSL) by 45.2%, style length (SL) by 16.4%, stigma brush (SB) by 25.7%, and stigma width (SW) by 38.6% compared to untreated control (T1) as an average of both seasons. Likewise, the evaluated CMS lines exhibited significant differences in floral traits. L2 displayed the uppermost SOA, DSO, TSL, SL, and SB, while L1 possessed the highest SW (Table 1). Otherwise, L5 recorded the lowest values of most of the evaluated floral traits. Furthermore, the evaluated CMS lines recorded different responses to the applied growth regulators. The maximum SOA, DSO, TSL, SL, and SB were assigned for L2 under foliar application of T3, while the highest SWwas recorded by L1 and L3 under T3. 

### 2.2. Growth-Related Traits

The results in Table 2 revealed that all growth-related traits were significantly (0.05) influenced by foliar application, CMS lines, and their interaction. The exogenous application of T3 with combination of GA_3_, IAA, and NAA displayed the highest values of the growth-related traits followed by T2. The application of T3 significantly enhanced flag leaf area (FLA) by 33.6%, plant height (PH) by 30.3%, panicle length (PL) by 17.0%, and panicle exertion (PE) by 52.7% compared to untreated control, as an average of both years. Moreover, the evaluated CMS lines reflected different growth behavior, L2 exhibited the remarkably higher values for the evaluated traits related to growth (FLA, PH, PL, and PE) than the other evaluated lines. Otherwise, L5 had the lowest values for FLA, PH, and PL, and L4 showed the lowest value for PE (Table 2). In addition, CMS lines responded differently to the application of growth regulators. The CMS L2 under foliar application of T3 exhibited the highest values for most growth traits. Otherwise, the lowest values for most traits were assigned for the untreated plants of L5 (Table 2). 

### 2.3. Yield-Related Traits

The analysis of variance implied a significant (0.05) influence of growth regulators, CMS lines, and their interaction on grain yield and its contributing traits (Table 3). The foliar application of T3 with combination of GA_3_, IAA, and NAA provided the uppermost values of grain yield (GY), fertile panicle (NFPH), seed set (SS), panicle weight (PW), and harvest index (HI), followed by T2 (Table 3). The exogenous application of T3 increased NFPH by 55.6%, PW by 108.6, SS by 127.5, GY by 151.5%, and HI by 49.4%, compared to untreated control T1 as an average of both seasons (Table 3). The CMS line L2 recorded the highest values for the evaluated yield traits compared to the other evaluated lines. Conversely, L5 displayed the lowest values for NFPH, PW, GY, and HI, and L4 showed the lowest SS. In terms of interaction between the foliar application and CMS lines, the CMS line L2 under foliar application T3 exhibited the highest values of all abovementioned yield traits (Table 3). Otherwise, untreated L5 showed the lowest values for NFPH, PW, SY, and HI, while untreated L4 had the lowest values for SS (Table 3).

### 2.4. Relationships among Treatments, CMS Lines, and Evaluated Traits

Multivariate analysis indicated that the first two PCs explained 88.12% of the total variation (Figure 1). The PC1 explained higher variation (73.90%) compared to PC2 and was related to foliar applications. The foliar applications were divided into three groups: T1 was situated on the negative side of PC1, T2 was located around the bi-plot origin, and T3 was located on the extremely positive side of the PC1. On the other hand, PC2 explained lower variation (14.22%) compared to PC1 and seemed to correspond with the CMS lines from top to bottom, L2, L1, L3, L4, and L5. The evaluated floral, growth, and yield traits were associated with T3 on the positive side of the PC1, particularly for CMS lines L2, L1, and L3. This indicates the high performance of floral traits, plant growth, and productivity of these lines under T3 compared to the other treatments. On the contrary, the untreated control T1 was on the opposite side, presenting lower floral traits, plant growth, and productivity of all evaluated CMS lines. The application of T2 was located between T1 and T3, indicating a slightly positive impact on the evaluated traits compared to untreated control. All floral, growth, and yield traits displayed strong positive inter-association by adjacent vectors except style length (SL), duration of spikelet opening (DSO), and spikelet opening angle (SOA), which had vectors with larger angles. The SL was located on the positive side of PC2 while DSO and SOA were on the negative side. This indicates a negative correlation between these traits, SL vs. DSO and SOA. Similarly, the heatmap and hierarchical clustering based on the measured traits divided the tested treatments (plant growth regulator × CMS line) into three main clusters (Figure 2). The plant growth regulators were the main dividing factor. Application T3 exhibited the highest values for most traits (depicted in red). On the contrary, application T1 had the lowest values (blue values), while T2 had intermediate values. In the cluster of T3, the CMS line L2 showed the highest values for most traits. Within the cluster of T1, the CMS line L5 seemed to have the lowest values for most traits. 

## 3. Discussion

The commercialization of hybrid rice is restricted by the low capacity of the CMS outcrossing rate. Enhancement of outcrossing ability is crucial to increase hybrid seed production. Plant growth regulators have valuable roles in plant development, plant growth, and productivity [23]. Several reports disclosed the importance of foliar application of plant growth regulators on rice seed production, but little is known about their impacts on floral traits. Floral characters considerably impact the efficacy of the outcrossing rate of CMS lines. Stimulation of floral traits in CMS lines increases the capacities of intercepting foreign pollens, consequently improving the cross-pollination process and hybrid seed production [10,24]. The present study exhibited that the foliar-applied gibberellic acid (GA_3_) solely (T2) or in combination with indole-3-acetic acid (IAA) and naphthalene acetic acid (NAA) (T3) remarkably improved floral traits compared to untreated control (T1). The combination of GA_3_, IAA, and NAA enabled all evaluated CMS lines to have a wider spikelet opening angle, longer duration of spikelet opening, longer stigma, longer style, higher stigma brush, and wider stigma. Increasing the duration of floret opening and the size of floret organs are favored features in outcrossing pollination. 

Poor panicle exertion of CMS lines is a serious limitation in hybrid seed production. In this respect, the exogenous application of GA_3_ uniquely or in combination with NAA and IAA exhibited a significant increase in panicle length and panicle exertion compared to untreated plants. Similar significant positive impacts of growth regulators on panicle length panicle exertion were disclosed by Abo-Youssef et al. [25], Hasan et al. [26], and Pin et al. [27]. Likewise, Huang et al. [28] assessed the impact of IAA; NAA; indole-3-butyric acid; 2,4-dichloro-phenoxy acetic acid; 3,6-dichloro-2-methoxy-benzoic acid; and abscisic acid on the floret closure of diverse rice genotypes. Their findings displayed that the applied growth regulators significantly increased the durations of floret opening while percentages of closed florets were lower in all evaluated rice genotypes. Furthermore, Kalavathi et al. [29] disclosed positive impacts of GA_3_ on panicle exertion and seed setting percentage. 

The auxins adjust turgor pressure in stomata guard cells and boost stomatal closure in plant leaves, hence elevating water uptake and cell expansion [28,30]. Consequently, floret closing in rice could be delayed by auxins due to their impacts on stimulating water uptake and preventing water loss in floret cells [28,30]. Furthermore, GA_3_ is a plant growth hormone that stimulates cell elongation and promotes the production of cellular mRNA, which ameliorates fast growth [31]. Moreover, IAA and NAA are plant hormones that enhance cell division, cell differentiation, cell elongation, protein synthesis, chlorophyll content, and photosynthetic efficiency [17,21]. Thus, foliar application by combining GA_3_, IAA, and NAA at the heading enhanced floret opening duration, stigma properties, panicle exertion, and rate of stigma exertion. 

The exogenously applied GA_3_, NAA, and IAA enhanced growth traits in all evaluated CMS lines. The applied combination of GA_3_, NAA, and IAA significantly enhanced flag leaf angle, plant height, panicle length, and panicle exertion. In this respect, Pan et al. [13] disclosed that the foliar application of growth regulators mitigates the impacts of rice senescence by modulating the activity of enzymatic antioxidants and enhancing the antioxidant system, which assisted in sustaining plant growth. Furthermore, the foliar application enhances root biomass and root activity, which improved phosphorus and potassium accumulation in rice stem and leaves and enhanced lodging resistance [13]. In this perspective, Hamad et al. [32], Pal et al. [33], and Hussain et al. [21] depicted that the foliar application of growth regulators enhanced plant height, panicle length, panicle exertion, and angle of flag leaf of rice. This improvement in vegetative growth reflected the highest number of fertile panicles per hill, panicle weight, seed set, grain yield, and harvest index. Similarly, Tiwari et al. [12], Biswas et al. [34], Hamad et al. [32], Pal et al. [33], and Hussain et al. [21] manifested that growth regulators boosted reproductive growth, seed set, and seed production rate in CMS lines. 

The evaluated CMS lines exhibited highly significant variations in the genetic behavior for floral, growth, and yield traits. L2 and L1 displayed the uppermost flag leaf angle, panicle length, plant height, panicle exertion, and yield traits compared to the other evaluated CMS lines, particularly under the T3 application. Similarly, Hashim et al. [10], Anis et al. [35], Maavimani and Saraswathi [36], and ElShamey et al. [6] detected genotypic differences among CMS lines in floral traits.

Understanding the association between the evaluated traits and treatments is an important aspect that can provide useful information [37,38,39]. The PC biplot and heatmap is an appropriate statistical approach to exploring the interrelationships among evaluated traits and treatments [40,41,42]. The PC biplot and heatmap have been convincingly employed to investigate the association among studied traits and treatments in various published reports [43,44,45]. The obtained results of the PCA biplot and heatmap reinforced the positive impacts of plant growth regulators on floral, growth, and yield traits. The PCA biplot and heatmap displayed that the evaluated floral, growth, and yield traits were associated with CMS lines L2 and L1 PC1, particularly under T3. These results confirmed that the application of T3 and using the CMS lines L2 and L1 enhances floral traits, plant growth, and rice productivity. Consequently, the exogenously applied GA_3_, NAA, and IAA in combination could be a beneficial tool in promoting floral traits, plant growth, and hybrid rice seed production in promising CMS lines. Furthermore, the PCA biplot revealed that all floral and growth traits were positively associated with yield traits. Consequently, enhancing these traits could improve the productivity of CMS lines and hybrid seed production. 

## 4. Materials and Methods

### 4.1. Experimental Site and Agricultural Treatments 

Field experimentation was performed at the Experimental Farm of Sakha Agricultural Research Station, Kafr El-Sheikh, Egypt (30°57′ N, 31°07′ E) during the summer of 2020 and 2021. The experimental site is characterized as a hot and arid climate with no precipitation during the summer season. The soil is an old Nile valley clay throughout the profile (12.6% sand, 32.4% silt, and 55% clay). Organic matter, pH, and electrical conductivity were 1.39 g/kg, 8.1, and 3.30 dS/m, respectively. Soluble cations and anions were 4.3, 1.88, 16, 4.55, and 5.55 mmolc/L for Mg^2+^, Na^+^, K^+^, Fe_3_+, and HCO^3−^, respectively. The experimental design was a split plot with three replications. The exogenous foliar applications of growth regulators were applied in the main plots, whereas the CMS lines were randomized in sub-plots. An isolation space of 100 m was considered for CMS seed production. Moreover, the experimental field was surrounded by an additional 20 rows of R lines to avoid any possibility of cross-pollination. Every main plot was isolated by a plastic barrier (2.5 m height) to avoid any pollen grain movement from one treatment to another. 

The nursery seedbed was well plowed and dry-leveled. Phosphorous fertilizer was applied at a rate of 50 kg P_2_O_5_/ha as super-phosphate (15.5% P_2_O_5_) and potassium at a rate of 45 kg K_2_O kg/ha as potassium sulfate (48% K_2_O) before tillage. Rice seeds at a rate of 20 kg/ha (15 kg of CMS line and 5 kg of the restorer line) were soaked in freshwater for 24 h, then drained and incubated for 48 h to hasten early germination. The seeds of each line were sown in the nursery on the 7 of May in both seasons. At the age of 30 days, seedlings were hand transplanted into hills with a row ratio of 2B:8A, 0.30 m between B line rows, 0.15 m between A line rows, 0.20 m between B and A lines, and 0.15 m between hills (A and B lines) with two seedlings per hill and each row was 5 m long. The row direction was perpendicular to the wind direction, and supplementary pollination was carried out artificially by shaking the pollen parent’s canopy with a stick during flowering to spread the pollen grains of A lines. The operation of shaking the pollen parent’s canopy was done three times between 9 and 11.30 a.m. at 30 min intervals for a period of 10 days. Nitrogen at the rate of 144 kg N/ha in the form of urea (46% N) was added in two splits: one-third as basal dressing and the rest at panicle initiation. 

### 4.2. Plant Material and Foliar Application

Five CMS lines were used in this study, including three from the International Rice Research Institute (IRRI) and Gambiaca and Kalinga from China, which were selected for their genetic differences (Table 4). Two foliar applications of growth regulators were applied comprising gibberellic acid (GA_3_, 300 g/ha) or GA_3_ (150 g/ha) + indole-3-acetic acid (IAA, 50 g/ha) + naphthalene acetic acid (NAA, 200 g/ha) versus untreated control. The applied plant growth regulators were purchased from Sigma. The growth regulators were applied in two foliar sprays after heading by three and six days as recommended by Tiwari et al. [12] and Pan et al. [21]. The growth regulators were dissolved in a tiny amount of 70% ethanol alcohol, combined with 50 L of water, and sprayed. 

### 4.3. Measured Traits

Data were recorded from five randomly selected hills excluding border rows per sub-plot for the studied characters. The floral traits were measured following Singh and Haque [46] using a micrometer under a stereomicroscope. The studied floral traits were duration of spikelet opening (min), spikelet opening angle (°), total stigma length (mm), style length (mm), stigma brush (mm), and stigma width (mm). Moreover, growth and yield traits were recorded for ten randomly selected plants from each plot. Plant height (cm), flag leaf angle (O), number of fertile tillers per hill, panicle length (cm), panicle exertion (%), panicle weight (g), seed set (%), grain yield (ton ha^−1^), and harvest index (%) were recorded. The crop was harvested when 85% of the grains became golden yellow. Grains were sun-dried and adjusted at 14% moisture content to estimate grain yield.
$$\mathrm{Panicle}\mathrm{exertion}\%=\frac{\mathrm{Length}\;\mathrm{of}\;\mathrm{exserted}\;\mathrm{part}\;\mathrm{of}\;\mathrm{panicle}\;\left(\mathrm{cm}\right)}{\mathrm{Panicle}\;\mathrm{length}\;\left(\mathrm{cm}\right)}\times 100\;\mathrm{Seed}\mathrm{set}\%=\frac{\mathrm{Number}\;\mathrm{of}\;\mathrm{filled}\;\mathrm{grains}/\mathrm{panicle}}{\mathrm{Total}\;\mathrm{number}\;\mathrm{of}\;\mathrm{spikelets}/\mathrm{panicle}}\;\times \;100\;$$

### 4.4. Statistical Analysis

R statistical software version 4.4 (Vienna, Austria) was applied to analyze the obtained data. Analysis of variance was performed for the split-plot design. The differences among foliar applications, CMS lines, and their interaction were separated by the Tukey HSD test at *p* ≤ 0.05. Biplot of a principal component analysis was performed, with ggplot2 package and the heatmap with RColorBrewer implemented in R software, to explore the relationship among evaluated traits.

## 5. Conclusions

Foliar application of gibberellic acid, indole-3-acetic acid, and naphthalene acetic acid in combination remarkably ameliorated all evaluated floral, growth, and yield characteristics of CMS lines compared to untreated plants. In addition, the evaluated CMS lines exhibited different genetic behavior for floral traits, plant growth, and plant productivity. L2 and L1 displayed the uppermost evaluated floral traits, plant growth, and yield traits, particularly under T3. Accordingly, it seems interesting to exploit foliar application of GA_3_, IAA, and NAA in combination as a useful tool in enhancing the outcrossing ability of promising CMS lines L2 and L1 to ameliorate outcrossing rates and hybrid seed production.

## Figures and Tables

**Figure 1 plants-11-01291-f001:**
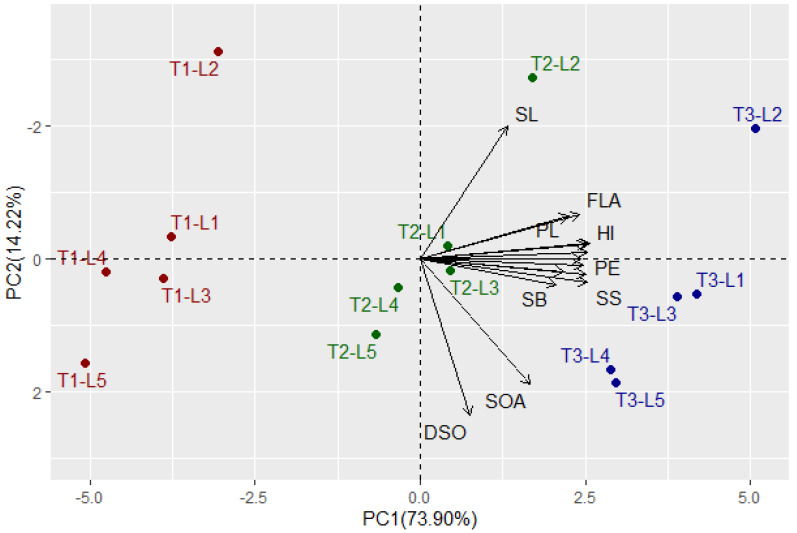
Principal component biplot for the applied growth regulators and CMS lines based on evaluated floral, growth, and yield traits over two growing seasons. The colors red, green, and blue indicate the foliar applications T1, T2, and T3, respectively. SOA: spikelet opening angle, DSO: duration of spikelet opening, TSL: total stigma length, SL: style length, SB: stigma brush, SW: stigma width, FLA: flag leaf angle, PH: plant height, PL: panicle length, PE: panicle exertion, NFPH: number of fertile panicles per hill, PW: panicle weight, SS: seed set, GY: grain yield, and HI: harvest index.

**Figure 2 plants-11-01291-f002:**
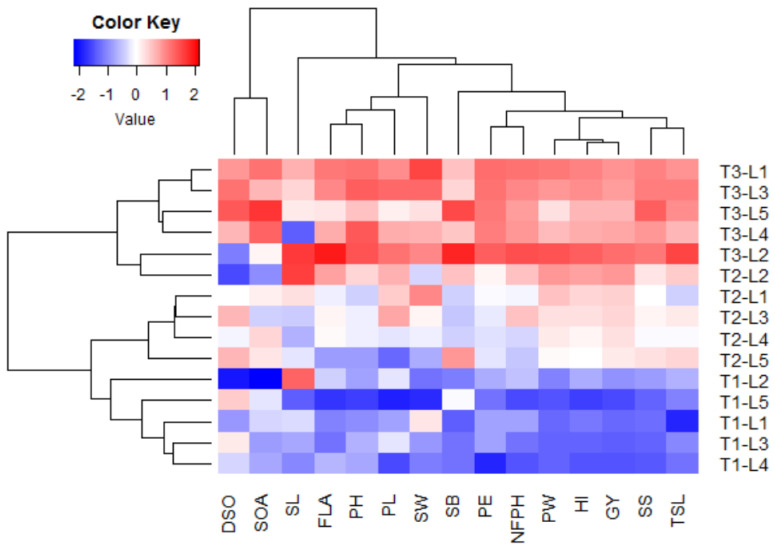
Heatmap and hierarchical clustering divide the growth regulators (T) and CMS lines (L) into different clusters based on floral, growth, and yield-related traits. Blue and red colors indicate low and high values for the corresponding trait, respectively. DSO: duration of spikelet opening, SOA: spikelet opening angle, SL: style length, FLA: flag leaf angle, TSL: total stigma length, SB: stigma brush, SW: stigma width, PH: plant height, PL: panicle length, PE: panicle exertion, NFPH: number of fertile panicles per hill, PW: panicle weight, SS: seed set, GY: grain yield, and HI: harvest index.

**Table 1 plants-11-01291-t001:** Influence of growth regulators on spikelet opening angle (SOA), duration of spikelet opening (DSO), total stigma length (TSL), style length (SL), stigma brush (SB), and stigma width (SW) of five diverse CMS lines.

Treatment	CMS Line	SOA (^°^)		DSO (min)		TSL (mm)		SL (mm)		SB (mm)		SW (mm)	
		2020	2021	2020	2021	2020	2021	2020	2021	2020	2021	2020	2021
**T1**	L1 (A1 × B1)	26.24 ^a^	27.56 ^a^	141.20 ^d^	143.83 ^d^	1.60 ^d^	1.50 ^d^	0.74 ^b^	0.75 ^b^	0.98 ^c^	1.08 ^c^	0.52 ^a^	0.56 ^a^
	L2 (A2 × B2)	27.02 ^a^	27.65 ^a^	172.00 ^a^	172.98 ^a^	2.03 ^a^	1.93 ^a^	0.96 ^a^	1.06 ^a^	1.22 ^a^	1.23 ^a^	0.40 ^c^	0.44 ^b^
	L3 (A3 × B3)	24.52 ^b^	25.78 ^b^	167.25 ^b^	166.22 ^b^	1.92 ^b^	1.82 ^b^	0.62 ^c^	0.72 ^c^	1.02 ^b^	1.12 ^b^	0.43 ^b^	0.47 ^b^
	L4 (A4 × B4)	25.54 ^b^	25.38 ^b^	154.00 ^c^	155.00 ^c^	1.84 ^c^	1.74 ^c^	0.58 ^d^	0.68 ^d^	1.01 ^b^	1.11 ^b^	0.41 ^bc^	0.45 ^b^
	L5 (A5 × B5)	20.00 ^c^	20.46 ^c^	116.26 ^e^	117.56 ^e^	1.83 ^c^	1.85 ^b^	0.53 ^e^	0.63 ^e^	1.01 ^b^	1.11 ^b^	0.35 ^d^	0.39 ^c^
	**Mean**	**24.66 ^C^**	**25.37 ^C^**	**150.14 ^C^**	**151.12 ^C^**	**1.84 ^C^**	**1.77 ^C^**	**0.69 ^C^**	**0.77 ^C^**	**1.05 ^C^**	**1.13 ^C^**	**0.42 ^C^**	**0.46 ^C^**
**T2**	L1 (A1 × B1)	28.52 ^a^	29.09 ^b^	162.67 ^h^	162.77 ^b^	2.13 ^d^	2.03 ^d^	0.80 ^b^	0.90 ^b^	1.12 ^c^	1.22 ^c^	0.59 ^a^	0.63 ^a^
	L2 (A2 × B2)	27.29 ^b^	31.04 ^a^	176.81 ^e^	176.91 ^a^	2.46 ^a^	2.36 ^a^	1.01 ^a^	1.11 ^a^	1.31 ^a^	1.41^a^	0.47 ^c^	0.51 ^c^
	L3 (A3 × B3)	26.49 ^b^	27.11 ^c^	176.72 ^e^	176.82 ^a^	2.43 ^a^	2.33 ^a^	0.71 ^c^	0.74 ^d^	1.26 ^b^	1.36 ^b^	0.51 ^b^	0.55 ^b^
	L4 (A4 × B4)	29.48 ^a^	29.75 ^b^	159.48 ^i^	160.00 ^b^	2.26 ^c^	2.16 ^c^	0.64 ^d^	0.74 ^d^	1.12 ^c^	1.22 ^c^	0.49 ^bc^	0.53 ^bc^
	L5 (A5 × B5)	24.18 ^c^	25.10 ^d^	127.39 ^m^	126.49 ^c^	2.36 ^b^	2.26 ^b^	0.71 ^c^	0.81 ^c^	1.11 ^c^	1.21 ^c^	0.44 ^d^	0.48 ^d^
	**Mean**	**27.19 ^B^**	**28.42 ^B^**	**160.61 ^B^**	**160.60 ^B^**	**2.33 ^B^**	**2.23 ^B^**	**0.77 ^B^**	**0.86 ^B^**	**1.18 ^B^**	**1.28 ^B^**	**0.50 ^B^**	**0.54 ^B^**
**T3**	L1 (A1 × B1)	32.19 ^b^	33.10 ^b^	185.26 ^c^	180.90 ^b^	2.63 ^c^	2.53 ^c^	0.86 ^b^	0.96 ^b^	1.26 ^c^	1.36 ^c^	0.64 ^a^	0.68 ^a^
	L2 (A2 × B2)	34.23 ^a^	35.04 ^a^	196.33 ^a^	192.07 ^a^	2.87 ^a^	2.77 ^a^	1.02 ^a^	1.12 ^a^	1.45 ^a^	1.55 ^a^	0.59 ^bc^	0.63 ^bc^
	L3 (A3 × B3)	30.17 ^c^	31.02 ^c^	188.67 ^b^	190.61 ^a^	2.69 ^b^	2.59 ^b^	0.81 ^c^	0.91 ^c^	1.40 ^b^	1.50 ^b^	0.61 ^ab^	0.65 ^ab^
	L4 (A4 × B4)	33.05 ^b^	33.38 ^b^	182.60 ^d^	171.12 ^c^	2.52 ^d^	2.42 ^d^	0.53 ^e^	0.63 ^e^	1.25 ^c^	1.35 ^c^	0.56 ^c^	0.60 ^c^
	L5 (A5 × B5)	28.20 ^d^	29.02 ^d^	137.70 ^l^	136.31 ^d^	2.65 ^c^	2.55 ^c^	0.78 ^d^	0.88 ^d^	1.23 ^d^	1.33 ^d^	0.53 ^d^	0.57 ^d^
	**Mean**	**31.57 ^A^**	**32.31 ^A^**	**178.11 ^A^**	**174.20 ^A^**	**2.67 ^A^**	**2.57 ^A^**	**0.80 ^A^**	**0.90 ^A^**	**1.32 ^A^**	**1.42 ^A^**	**0.59 ^A^**	**0.63 ^A^**
**ANOVA**													
CMS line (L)		<0.001	<0.001	<0.001	<0.001	<0.001	<0.001	<0.001	<0.001	<0.001	<0.001	<0.001	<0.001
Treatment (T)		<0.001	<0.001	<0.001	<0.001	<0.001	<0.001	<0.001	<0.001	<0.001	<0.001	<0.001	<0.001
G × L		0.002	0.001	<0.001	<0.001	<0.001	<0.001	<0.001	<0.001	<0.001	<0.001	0.011	0.011

T1: untreated control, T2: gibberellic acid (GA_3_, 300 g/ha), and T3: GA_3_ (150 g/ha) + indole-3-acetic acid (IAA, 50 g/ha) + naphthalene acetic acid (NAA, 200 g/ha). Different lowercase letter within a column indicates significant difference among the CMS lines within each growth regulator at *p* < 0.05. Different uppercase letter within a column indicates significant difference among growth regulator means at *p* < 0.05.

**Table 2 plants-11-01291-t002:** Influence of growth regulators on flag leaf angle (FLA), plant height (PH), panicle length (PL), and panicle exertion (PE) of five diverse CMS lines.

Treatment	CMS Line	FLA (^°^)		PH (cm)		PL (cm)		PE (%)	
		2020	2021	2020	2021	2020	2021	2020	2021
**T1**	L1 (A1 × B1)	25.91 ^c^	26.02 ^b^	88.34 ^b^	90.33 ^b^	20.14 ^b^	20.93 ^ab^	54.14 ^a^	54.98 ^a^
	L2 (A2 × B2)	28.70 ^a^	29.02 ^a^	90.25 ^ab^	92.17 ^ab^	21.88 ^b^	21.28 ^a^	55.20 ^a^	55.54 ^a^
	L3 (A3 × B3)	24.50 ^d^	26.16 ^b^	92.59 ^a^	93.51 ^a^	23.08 ^a^	20.10 ^bc^	53.94 ^a^	54.40 ^a^
	L4 (A4 × B4)	27.24 ^b^	28.27 ^ab^	90.14 ^ab^	93.61 ^a^	18.52 ^c^	19.68 ^cd^	40.79 ^c^	43.22 ^c^
	L5 (A5 × B5)	22.46 ^e^	23.70 ^c^	78.85 ^c^	83.89 ^c^	18.13 ^c^	18.65 ^d^	49.09 ^b^	50.71 ^b^
	**Mean**	**25.76 ^C^**	**26.64 ^C^**	**88.03 ^C^**	**90.70 ^C^**	**20.35 ^C^**	**20.13 ^B^**	**50.63 ^C^**	**51.77 ^C^**
**T2**	L1 (A1 × B1)	28.86 ^c^	30.81 ^c^	95.97 ^c^	96.98 ^c^	22.43 ^ab^	23.46 ^bc^	60.54 ^b^	65.20 ^ab^
	L2 (A2 × B2)	33.65 ^a^	34.51 ^a^	104.79 ^a^	107.64 ^a^	23.56 ^a^	23.12 ^a^	63.95 ^a^	66.65 ^a^
	L3 (A3 × B3)	30.71 ^b^	31.59 ^b^	98.70 ^b^	100.16 ^b^	23.59 ^a^	23.31 ^a^	58.45 ^c^	64.26 ^b^
	L4 (A4 × B4)	30.60 ^b^	31.22 ^b^	98.43 ^b^	100.90 ^b^	20.60 ^bc^	22.48 ^cd^	58.80 ^c^	62.76 ^c^
	L5 (A5 × B5)	26.50 ^d^	27.29 ^d^	89.62 ^d^	92.44 ^d^	18.83 ^c^	20.37 ^d^	60.57 ^b^	61.46 ^c^
	**Mean**	**30.06 ^B^**	**31.08 ^B^**	**97.50 ^B^**	**99.62 ^B^**	**21.80 ^B^**	**22.55 ^A^**	**60.46 ^B^**	**64.07 ^B^**
**T3**	L1 (A1 × B1)	35.09 ^b^	36.02 ^b^	115.54 ^b^	117.60 ^b^	23.70 ^b^	24.18 ^a^	77.88 ^b^	78.85 ^ab^
	L2 (A2 × B2)	39.15 ^a^	38.80 ^a^	118.34 ^a^	121.12 ^a^	24.05 ^ab^	24.54 ^a^	79.96 ^a^	80.45 ^a^
	L3 (A3 × B3)	34.78 ^b^	35.52 ^b^	118.26 ^a^	119.08 ^ab^	25.02 ^a^	24.00 ^a^	77.67 ^b^	78.28 ^b^
	L4 (A4 × B4)	33.47 ^c^	34.10 ^c^	119.81 ^a^	118.42 ^ab^	23.04 ^b^	23.76 ^ab^	77.12 ^b^	76.75 ^b^
	L5 (A5 × B5)	30.83 ^d^	32.36 ^d^	107.83 ^c^	108.76 ^c^	21.91 ^c^	22.66 ^b^	77.59 ^b^	77.08 ^b^
	**Mean**	**34.66 ^A^**	**35.36 ^A^**	**115.95 ^A^**	**117.00 ^A^**	**23.54 ^A^**	**23.83 ^A^**	**78.04 ^A^**	**78.28 ^A^**
**ANOVA**									
CMS line (L)		<0.001	<0.001	<0.001	<0.001	<0.001	0.009	<0.001	<0.001
Treatment (T)		<0.001	<0.001	<0.001	<0.001	0.005	0.015	<0.001	<0.001
L × T		0.004	0.003	0.002	0.009	0.045	0.039	<0.001	<0.001

T1: untreated control, T2: gibberellic acid (GA_3_, 300 g/ha), and T3: GA_3_ (150 g/ha) + indole-3-acetic acid (IAA, 50 g/ha) + naphthalene acetic acid (NAA, 200 g/ha). Different lowercase letter within a column indicates significant difference among the CMS lines within each growth regulator at *p* < 0.05. Different uppercase letter within a column indicates significant difference among growth regulator means at *p* < 0.05.

**Table 3 plants-11-01291-t003:** Influence of growth regulators on number of fertile panicles per hill (NFPH), panicle weight (PW), seed set (SS), grain yield (GY), and harvest index (HI) of five diverse CMS lines.

Treatment	CMS Line	NFPH		PW (g)		SS (%)		GY (ton/ha)		HI (%)	
		2020	2021	2020	2021	2020	2021	2020	2021	2020	2021
**T1**	L1 (A1 × B1)	10.34 ^a^	12.34 ^a^	1.20 ^b^	1.25 ^a^	10.80 ^b^	12.36 ^b^	0.65 ^b^	0.71 ^b^	12.11 ^ab^	13.02 ^b^
	L2 (A2 × B2)	11.01 ^a^	13.01 ^a^	1.30 ^a^	1.42 ^b^	13.41 ^a^	14.95 ^a^	0.78 ^a^	0.86 ^a^	13.09 ^a^	14.34 ^a^
	L3 (A3 × B3)	9.26 ^b^	11.26 ^b^	1.18 ^bc^	1.22 ^c^	10.32 ^b^	11.97 ^bc^	0.62 ^c^	0.65 ^c^	11.33 ^b^	12.76 ^b^
	L4 (A4 × B4)	8.67 ^b^	10.67 ^b^	1.16 ^c^	1.21 ^c^	9.65 ^b^	11.34 ^c^	0.58 ^d^	0.60 ^d^	11.01 ^b^	12.43 ^b^
	L5 (A5 × B5)	8.35 ^b^	10.35 ^b^	1.06 ^d^	1.18 ^c^	9.98 ^b^	11.91 ^bc^	0.55 ^e^	0.57 ^e^	10.90 ^b^	11.65 ^c^
	**Mean**	**9.53 ^C^**	**11.53 ^C^**	**1.18 ^C^**	**1.26 ^C^**	**10.83 ^C^**	**12.51 ^C^**	**0.64 ^C^**	**0.68 ^C^**	**11.69 ^C^**	**12.84 ^C^**
**T2**	L1 (A1 × B1)	12.56 ^b^	13.89 ^b^	2.30 ^b^	2.36 ^b^	18.62 ^c^	20.31 ^c^	1.40 ^b^	1.49 ^b^	16.36 ^ab^	17.12 ^b^
	L2 (A2 × B2)	13.96 ^a^	15.96 ^a^	2.54 ^a^	2.58 ^a^	20.22 ^a^	21.86 ^ab^	1.63 ^a^	1.72 ^a^	17.81 ^a^	18.09 ^a^
	L3 (A3 × B3)	13.90 ^a^	15.90 ^a^	2.18 ^c^	2.17 ^c^	19.14 ^b^	20.98 ^bc^	1.39 ^c^	1.46 ^c^	16.04 ^ab^	16.84 ^bc^
	L4 (A4 × B4)	11.52 ^c^	13.52 ^b^	2.11 ^d^	2.14 ^c^	18.19 ^c^	20.06 ^c^	1.35 ^d^	1.41 ^d^	15.81 ^b^	16.22 ^cd^
	L5 (A5 × B5)	11.20 ^c^	13.20 ^b^	2.07 ^e^	2.03 ^d^	20.26 ^a^	22.43 ^a^	1.30 ^e^	1.39 ^e^	15.21 ^b^	16.01 ^d^
	**Mean**	**12.63 ^B^**	**14.50 ^B^**	**2.24 ^B^**	**2.26 ^B^**	**19.29 ^B^**	**21.13 ^B^**	**1.41 ^B^**	**1.49 ^B^**	**16.25 ^B^**	**16.86 ^B^**
**T3**	L1 (A1 × B1)	15.70 ^ab^	17.70 ^ab^	2.60 ^b^	2.84 ^b^	24.11 ^c^	28.17 ^b^	1.65 ^b^	1.71 ^b^	18.16 ^ab^	19.22 ^ab^
	L2 (A2 × B2)	16.50 ^a^	18.50 ^a^	2.90 ^a^	2.92 ^a^	26.51 ^a^	29.74 ^a^	1.81 ^a^	1.85 ^a^	19.14 ^a^	19.76 ^a^
	L3 (A3 × B3)	15.18 ^bc^	17.18 ^b^	2.50 ^c^	2.60 ^c^	25.11 ^b^	27.79 ^b^	1.62 ^c^	1.68 ^c^	17.88 ^ab^	18.87 ^bc^
	L4 (A4 × B4)	14.81 ^c^	16.81 ^b^	2.20 ^d^	2.51 ^d^	23.89 ^c^	26.66 ^c^	1.56 ^d^	1.63 ^d^	17.02 ^b^	18.34 ^cd^
	L5 (A5 × B5)	14.71^c^	16.71 ^b^	2.15 ^e^	2.23 ^e^	25.34 ^b^	28.15 ^b^	1.51 ^e^	1.58 ^e^	16.82 ^b^	18.03 ^d^
	**Mean**	**15.38 ^A^**	**17.38 ^A^**	**2.47 ^A^**	**2.62 ^A^**	**24.99 ^A^**	**28.10 ^A^**	**1.63 ^A^**	**1.69 ^A^**	**17.80 ^A^**	**18.84 ^A^**
**ANOVA**											
CMS line (L)		<0.001	<0.001	<0.001	<0.001	<0.001	<0.001	<0.001	<0.001	<0.001	<0.001
Treatment (T)		<0.001	<0.001	<0.001	<0.001	<0.001	<0.001	<0.001	<0.001	<0.001	<0.001
L × T		0.022	0.011	<0.001	<0.001	<0.001	<0.001	<0.001	<0.001	0.045	0.043

T1: untreated control, T2: gibberellic acid (GA_3_, 300 g/ha), and T3: GA_3_ (150 g/ha) + indole-3-acetic acid (IAA, 50 g/ha) + naphthalene acetic acid (NAA, 200 g/ha). Different lowercase letter within a column indicates significant difference among the CMS lines within each growth regulator at *p* < 0.05. Different uppercase letter within a column indicates significant difference among growth regulator means at *p* < 0.05.

**Table 4 plants-11-01291-t004:** Cytoplasmic male sterile (CMS) lines used for the study.

Code	CMS Line	CMS Line Code	Maintainer	Maintainer Code	Days to Heading	Cytoplasmic Source	Origin
L1	IR69625A	A1	IR69625B	B1	104.5	Wild abortive (WA) CMS line	IRRI
L2	IR58025A	A2	IR58025B	B2	108.2	Wild abortive (WA) CMS line	IRRI
L3	IR70368A	A3	IR70368B	B3	103.2	Wild abortive (WA) CMS line	IRRI
L4	G46A	A4	G46B	B4	88.9	Gambiaca CMS line	China
L5	K17A	A5	K17B	B5	84.5	Kalinga type	China

## Data Availability

The data presented in this study are available upon request from the corresponding author.

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
