# Peer review of "Growth Regulators Improve Outcrossing Rate of Diverse Rice Cytoplasmic Male Sterile Lines through Affecting Floral Traits"

_plants, 2022, doi:10.3390/plants11101291_

Round 1

Reviewer 1 Report

The authors of the manuscript ‘Growth Regulators Ameliorates Floral Traits and Outcrossing  Rate of Diverse Rice Cytoplasmic Male Sterile Lines’ examined the effect of exogenous application of plant growth regulators on the outcrossing rate in rice cytoplasmic male sterile lines. The research presented is routine rather than a novel. The manuscript is well prepared, and the results are encouraging. Please indicate the novelty of this study.

L39: Please correct: L3, L3, and L1.

L107: Figure 1. Please introduce a scale bar.

L253-255: Two foliar applications of growth regulators were applied comprising gibberellic acid (GA3, 300 g/ha) or GA3 (150 g/ha) + indole-3-acetic acid (IAA, 50 254 g/ha) + naphthalene acetic acid (NAA, 200 g/ha). What is the rationale for choosing the above concentrations of plant growth regulators?

L254-255: Please indicate the source of gibberellic acid, indole-3-acetic acid, and naphthalene acetic acid. They are commercial or laboratory grades.

L255-256: The growth regulators were applied in two foliar sprays after heading by three and six days. Based on previous studies?

Author Response

Responses to Reviewers' Comments

Reviewer 1:

The authors of the manuscript ‘Growth Regulators Ameliorates Floral Traits and Outcrossing  Rate of Diverse Rice Cytoplasmic Male Sterile Lines’ examined the effect of exogenous application of plant growth regulators on the outcrossing rate in rice cytoplasmic male sterile lines. The research presented is routine rather than a novel. The manuscript is well prepared, and the results are encouraging. Please indicate the novelty of this study.

Re: We would like to thank the Reviewer for his time dedicated to our manuscript and his/her positive assessment of our work. Yes, several earlier reports elucidated the impact of exogenous application of plant growth regulators on rice seed production, but little is known about their impacts on floral traits. Regulation of floral traits is vitally decisive in hybrid rice seed production. Accordingly, in the present study, we assessed the impact of the combination of GA3, IAA, and NAA on floret opening duration, stigma properties, panicle exertion, and their impact on the outcrossing rate of diverse five male sterile lines. Thanks for the suggestion, this has been clarified in the introduction and discussion sections please see lines 65-77 and 228-231.

L39: Please correct: L3, L3, and L1.

Re: “L3, L3, and L1” has been corrected to be “L2, L3, and L1” (line 44)

L107: Figure 1. Please introduce a scale bar.

Re: Scale bar has been added and the Figure has been moved to the supplementary materials

L253-255: Two foliar applications of growth regulators were applied comprising gibberellic acid (GA3, 300 g/ha) or GA3 (150 g/ha) + indole-3-acetic acid (IAA, 50 254 g/ha) + naphthalene acetic acid (NAA, 200 g/ha). What is the rationale for choosing the above concentrations of plant growth regulators?

Re: The used plant growth regulators and their concentrations were applied based on their positive impacts on plant vegetative and productivity growth in earlier published reports more details have been added please see lines 356-357.

L254-255: Please indicate the source of gibberellic acid, indole-3-acetic acid, and naphthalene acetic acid. They are commercial or laboratory grades.

Re: They are commercial and were purchased from Sigma, more details have been added please see lines 354-355.

L255-256: The growth regulators were applied in two foliar sprays after heading by three and six days. Based on previous studies?

Re: Yes, the growth regulators were applied based on the previous studies, more details have been added (lines 356-357).

Reviewer 2 Report

The overall impression from this paper is that the authors were not aware why this particular study was necessary, and what are the relevance of the obtained results in the context of scientific knowledge. All major effects described in this particular study have been described already numerous times for the same model plant.

Title

The phrase "ameliorates floral traits" seems to be somehow inaccurate, as "traits" cannot be "ameliorated" but expression of these traits can. May be "improve" is a better term? In addition, "outcrossing rate" is the desired result, therefore, it seems that it is better to express like this: "Growth regulators improve outcrossing rate of diverse rice cytoplasmic male sterile lines through affecting floral traits" or similar.

Abstract

"Enhancing floral traits" could be replaced by "improving floral traits" (line 24).

Combine 2nd and 3rd sentence (lines 24–27).

Introduce abbreviations right after first mention of the terms (lines 27–28).

Introduction

Some references are not correctly used, not pointing to experimental papers where the particular fact has been found or review papers describing the particular relationship. It is likely that only some idea from Introduction of the referenced papers was taken. This concerns at least the following references:

  • line 47 [1],
  • line 49 [3],
  • line 66 [13] is out of the context as it is not about rice,
  • line 71 [15,16],

It needs to be explained what is a scientific relevance of this particular study, as very similar studies have been performed, as [32] already 10 years ago. Information on previous similar studies always needs to be provided in Introduction. Also, aim of the study needs to be more specifically formulated.

Results

Table 1 seems to be redundant, as these are averaged date by treatments and by lines, further reported by line vs. treatment combinations in Table 2. Statistical significance of interaction as resulted from ANOVA can be reported in the text. Photographs in Fig. 1 are of low scientific relevance, as individual stigmas are shown instead of complex picture, without scale bars, and some of them seems to be magnified. Tables 3 and 5 are also redundant for the same reason as for Table 1.

In general, the results in 2.1, 2.2 and 2.3 need to be described in more detail.

In 2.4, instead of using PCA, consider using other types of multivariate analysis, as heatmaps together with cluster analysis, which could give better insight in functional relationships among different traits and relatedness between genotypes.

Discussion

This part is extremely limited. Relevance of the particular study needs to be clearly pointed out. Pointing only to number of studies with similar results do not provide any clue for necessity of the performed study.

At least, some functional analysis of particular effects need to be provided instead of repeating of general texts on effects of these growth regulators. For example, why combination treatment could give better results etc.

It is also important that any use of plant growth regulators in agricultural practice can be beneficial only in the case if optimum mineral nutrition is provided in turn ensuring efficient photosynthesis as a functional basis for any growth enhancements. Otherwise it might seem that growth regulators itself provide necessary resources.

Materials and methods

No information provided on production of seedlings for experiment. Plot size and number of seedlings per plot not specified. Timing of treatments, artificial pollination, harvest etc. not clear.

Author Response

Responses to Reviewers' Comments

Reviewer 2:

The overall impression from this paper is that the authors were not aware why this particular study was necessary, and what are the relevance of the obtained results in the context of scientific knowledge. All major effects described in this particular study have been described already numerous times for the same model plant.

Re: We would like to thank the Reviewer for providing constructive criticism to improve the quality of the manuscript. Yes, you are right several earlier reports elucidated the impact of exogenous application of plant growth regulators on rice seed production, but little is known about their impacts on floral traits. Regulation of floral traits is vitally decisive in hybrid rice seed production. Accordingly, in the present study, we assessed the impact of the combination of GA3, IAA, and NAA on floret opening duration, stigma properties, panicle exertion, and their impact on the outcrossing rate of diverse five male sterile lines. The importance of the studied topic has been improved in the introduction, and the discussion section and its relevance with earlier studies have been presented please see lines 65-77 and 228-231.

Title

The phrase "ameliorates floral traits" seems to be somehow inaccurate, as "traits" cannot be "ameliorated" but expression of these traits can. Maybe "improve" is a better term? In addition, "outcrossing rate" is the desired result, therefore, it seems that it is better to express like this: "Growth regulators improve outcrossing rate of diverse rice cytoplasmic male sterile lines through affecting floral traits" or similar.

Re: The title has been modified as suggested

Abstract

"Enhancing floral traits" could be replaced by "improving floral traits" (line 24).

Re: This sentence has been combined with the following sentence as suggested by the reviewer and the word “enhance” has been replaced by “improve” as recommended (lines 25-28 in the revised version)

Combine 2nd and 3rd sentence (lines 24–27).

Re: Both sentences have been combined as suggested (lines 26-29)

Introduce abbreviations right after first mention of the terms (lines 27–28).

Re: The abbreviations have been introduced after the first mention of the terms (lines 32-33 in the revised version)

 Introduction

Some references are not correctly used, not pointing to experimental papers where the particular fact has been found or review papers describing the particular relationship. It is likely that only some idea from Introduction of the referenced papers was taken. This concerns at least the following references: line 47 [1], line 49 [3], line 66 [13] is out of the context as it is not about rice, line 71 [15,16],

Re: All references have been revised and modified as suggested

It needs to be explained what is a scientific relevance of this particular study, as very similar studies have been performed, as [32] already 10 years ago. Information on previous similar studies always needs to be provided in Introduction. Also, aim of the study needs to be more specifically formulated.

Re: We highly appreciate the suggestions provided by the reviewer. The scientific relevance of this study and its association with earlier studies have been explained in the introduction (please see lines 65-77) and the objectives have been revised and improved (lines 91-98).

Results

Table 1 seems to be redundant, as these are averaged date by treatments and by lines, further reported by line vs. treatment combinations in Table 2. Statistical significance of interaction as resulted from ANOVA can be reported in the text. Photographs in Fig. 1 are of low scientific relevance, as individual stigmas are shown instead of complex picture, without scale bars, and some of them seems to be magnified. Tables 3 and 5 are also redundant for the same reason as for Table 1.

Re: Tables 1, 3, and 5 have been deleted, and the averages of the main effect of growth regulators and ANOVA results have been combined in Tables 2, 4, and 6. The Figure 1, a scale bar has been and has been moved to the supplementary materials

In general, the results in 2.1, 2.2 and 2.3 need to be described in more detail.

Re: The mentioned sections have been revised and improved as suggested

In 2.4, instead of using PCA, consider using other types of multivariate analysis, as heatmaps together with cluster analysis, which could give better insight in functional relationships among different traits and relatedness between genotypes.

Re: Heatmap together with cluster analysis have been performed and added to the manuscript as suggested, please see lines 214-223.

 Discussion

This part is extremely limited. Relevance of the particular study needs to be clearly pointed out. Pointing only to number of studies with similar results do not provide any clue for necessity of the performed study.

Re: Thanks for the suggestion, the discussion has been rewritten and considerably improved. The relevance of the study has been discussed better please see lines 228-241.

At least, some functional analysis of particular effects need to be provided instead of repeating of general texts on effects of these growth regulators. For example, why combination treatment could give better results etc.

Re: Thanks for the suggestion, more explanations have been added to the discussion please see lines 253-260.  

It is also important that any use of plant growth regulators in agricultural practice can be beneficial only in the case if optimum mineral nutrition is provided in turn ensuring efficient photosynthesis as a functional basis for any growth enhancements. Otherwise, it might seem that growth regulators itself provide necessary resources.

Re: Thanks for the suggestions, more explanations have been added, please see lines 270-276.

Materials and methods

No information provided on production of seedlings for experiment. Plot size and number of seedlings per plot not specified. Timing of treatments, artificial pollination, harvest etc. not clear.

Re:  The required details have been added to the materials and methods, please see lines 335-348.

Round 2

Reviewer 2 Report

All remarks has been addressed. Most importantly, both introduction and discussion, as well as results part have been improved.